# Self-Refining Diffusion Samplers: Enabling Parallelization via Parareal Iterations

**Nikil Roashan Selvam**     **Amil Merchant**     **Stefano Ermon**
Department of Computer Science
Stanford University
{nrs,amil,ermon}@cs.stanford.edu

## Abstract

In diffusion models, samples are generated through an iterative refinement process, requiring hundreds of sequential model evaluations. Several recent methods have introduced approximations (fewer discretization steps or distillation) to trade off speed at the cost of sample quality. In contrast, we introduce Self-Refining Diffusion Samplers (SRDS) that retain sample quality and can improve latency at the cost of additional parallel compute. We take inspiration from the Parareal algorithm, a popular numerical method for parallel-in-time integration of differential equations. In SRDS, a quick but rough estimate of a sample is first created and then iteratively refined *in parallel* through Parareal iterations. SRDS is not only guaranteed to accurately solve the ODE and converge to the serial solution but also benefits from parallelization across the diffusion trajectory, enabling batched inference and pipelining. As we demonstrate for pre-trained diffusion models, the early convergence of this refinement procedure drastically reduces the number of steps required to produce a sample, speeding up generation for instance by up to 1.7x on a 25-step StableDiffusion-v2 benchmark and up to 4.3x on longer trajectories.

## 1 Introduction

Deep generative models based on diffusion processes have showcased the capability to produce high-fidelity samples in a wide-range of applications [28, 11, 39, 27]. From their origins in image and audio generation [31, 33, 10], diffusion models have enabled robotic applications as well as scientific discovery via drug design [1]. Despite this promise, sampling from diffusion models can still be prohibitively slow. Early Denoising Diffusion Probabilistic Models [10] required a thousand sequential model evaluations (steps), and state-of-the-art models such as StableDiffusion [27] can still require up to 100s of iterations for high-quality generations. This large number of sampling steps leads to high-latencies associated with diffusion models, limiting applications such as real-time image or music editing and trajectory planning in robotics [13, 12].

As sampling involves solving an ordinary differential equation (ODE), a prominent body of research — including works such as Denoising Diffusion Implicit Models (DDIM, [32]), Diffusion Exponential Integrator Sampler (DEIS, [42]), and DPM-Solver [19] — has tried to reduce the number of model evaluations required by introducing various approximations. For example, progressive distillation [29] requires re-training models to approximate the solution to the ODE at larger timesteps. However, such approaches trade-off speed at the cost of sample quality.

In this work, we instead take an orthogonal approach: we focus on additional **parallel** compute and show how this can be used to reduce latencies while still providing accurate solutions to the original ODE, thereby preserving sample quality. Recently Shih et al. [30] leveraged a parallel-in-time integration method to introduce the first highly-parallelizable algorithm for diffusion model sampling. The presented ParaDiGMs algorithm builds on Picard iterations [26] to perform denoising steps

38th Conference on Neural Information Processing Systems (NeurIPS 2024).

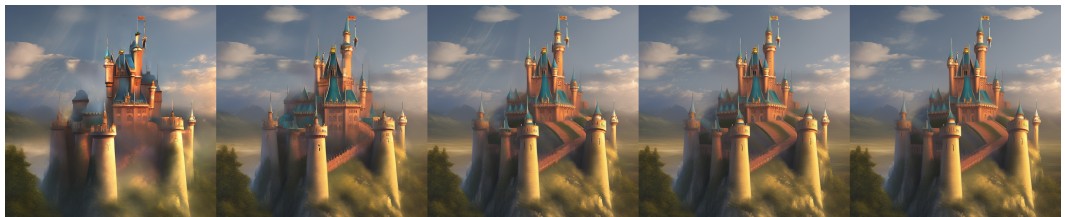

Figure 1: A visualization of the iterative refinement provided by the SRDS algorithm on a sample from StableDiffusion with the prompt 'a beautiful castle, matte painting.' The initial coarse solve (left) via limited steps provides a rough estimate of the sample, which iteratively refined through iterations of our algorithm. Early convergence is observed as the 3rd output nearly matches, a key feature that enables efficient generation.

across the trajectory in parallel, leading to state-of-the-art sampling speeds on popular benchmarks [30]. Despite the promising results, ParaDiGMs can be highly memory bound due to the use of sliding window methods and also has a sequential convergence criteria requiring communication-expensive cumulative sums across devices to coordinate parallel-sampling. The method also has limited controllability over the accuracy of the final solution, assuming convergence per step rather than in the final generation. It is also worth noting that in concurrent work, Tang et al. [37] take a completely different approach to accelerating parallel sampling: special techniques in solving triangular nonlinear systems through fixed point iteration. Instead, in this paper, we turn to multiple-shooting methods from the parallel-in-time ODE integration literature [14, 3, 6] and aim to improve parallelization of diffusion sampling by utilizing multiple resolutions across the time domain [21].

Specifically, we present **Self-Refining Diffusion Samplers (**SRDS**)** that start with a quick but rough solve of the diffusion trajectory, achieved by limiting the number of total steps taken (for instance, using a few-step DDIM solver). The trajectory can then be simulated to higher fidelity via a highly-parallel algorithm that updates the final generation iteratively until convergence. At a high level, each refinement step of SRDS partitions the current guess of the trajectory into blocks, and simulates each of these blocks at the desired (high) resolution. The running guess for the overall trajectory is then updated via a predictor-corrector mechanism based on the Parareal algorithm to accelerate convergence. This iterative refinement allows us to efficiently interpolate *in parallel* between a coarse solution corresponding to a low-fidelity sample and an accurate solution corresponding to a high-fidelity sample. The key benefits of SRDS are three-fold:

**Approximation-Free:** By design, SRDS computes an accurate solution to the reverse process (as defined by the practitioner's choice of diffusion solver), thereby maintaining high quality of samples. Importantly, as it is purely a sampling algorithm, it does so without incurring any retraining cost.

**Extensive Control and Compatibility:** By serving as an efficient interpolation method between the coarse and fine-grained solvers, SRDS provides the practitioner with flexible control of the tradeoff between sample quality and speed. For instance, one could first start with a rough solve (corresponding to, say, few-step DDIM). Then, if desired, one can add a budget-appropriate number of *parallel* SRDS iterations (instead of sequential) to refine the obtained sample. Furthermore, SRDS is compatible with most existing off-the-shelf solvers (such as Euler, Heun, DPM etc), thereby providing direct benefits to virtually any diffusion workflow.

**Low Latency:** Most importantly, we find that the number of iterations required by SRDS for convergence is quite low, leading to drastic improvements in latency for sampling. We present results using the Self-Refining Diffusion Samplers on a wide range of benchmarks, starting with pixel-based image diffusion models and further exploring latent-methods where SRDS leads to a 1.7x improvement in the sampling speed on the 25-step StableDiffusion-v2 benchmark and up to 4.3x on longer trajectories. [1] Through enabling faster sampling, SRDS aims to unlock capabilities for real-time interaction with diffusion models.

---

[1] Code for our paper can be found at `https://github.com/nikilrselvam/srds`.

## 2 Background

Diffusion models are a general class of generative models that rely on a noising procedure that converts the data distribution into noise via a series of latent variables updates. For the purposes of this work, we will consider the continuous-time generalization presented by Song [35] and Denoising Diffusion Implicit Models [10] that formulate sampling as solving the initial value problem characterized by the probability flow ordinary differential equation (ODE):

$$d\boldsymbol{x} = \underbrace{\left[ \boldsymbol{f}(\boldsymbol{x},t) - \frac{1}{2}g(t)^2 s_\theta(\boldsymbol{x},t) \right]}_{h_\theta} dt; \quad \boldsymbol{x}(t=0) = \boldsymbol{x}_0 \sim \mathcal{N}(\boldsymbol{0}, \boldsymbol{I}) \tag{1}$$

where $s_\theta(x,t)$ is a time-conditional prediction of the score function $\nabla_x \log p_t(\boldsymbol{x})$ from the diffusion model. To be consistent with prior work on parallelized diffusion sampling Shih et al. [30], we use a reversed time index (from traditional notation) where $\boldsymbol{x}_0$ refers to pure Gaussian noise, and $\boldsymbol{x}_T$ refers to the denoised image after $T$ denoising steps.

### 2.1 Solving the Differential Equation

Given the dynamics governing the differential equation, our goal is to provide accurate solutions to:

$$\boldsymbol{x}_T = \boldsymbol{x}_0 + \int_{t=0}^{T} h_\theta(\boldsymbol{x},t)dt \tag{2}$$

in order to produce a sample from the diffusion model. Common approaches discretize the time interval $[0, T]$ into $N$ pieces ($t_0{=}0, t_1, t_2, ..., t_N{=}T$) and solve a sequence of initial value problems to yield an approximation $(\boldsymbol{x}_0, \boldsymbol{x}_1, ..., \boldsymbol{x}_N{=}\boldsymbol{x}_T)$ to the trajectory.

Formally, a *solver* is a function $\mathcal{F}(\boldsymbol{x}_{start}, t_{start}, t_{end})$ that propagates $x$ from $t = t_{start}$ with initial value $\boldsymbol{x}_{start}$ to $t = t_{end}$. Solving the differential equation corresponds to approximating the solution $\boldsymbol{x}_T$ to the given initial value problem by a sequence of $N$ solves:

$$\boldsymbol{x}_{i+1} = \mathcal{F}(\boldsymbol{x}_i, t_i, t_{i+1}) \ \ \forall i \in [0, N-1]; \quad \text{given initial value } x_0 \tag{3}$$

The choice of solver $\mathcal{F}$ dictates the sampling speed and accuracy of the solution. In practice, solvers which are accurate are often slow (due to high number of evaluations of $h_\theta$), whereas solvers that are fast tend to have reduced accuracy. Initial works on diffusion models used the classical Euler method as choice of $\mathcal{F}$, and it can be expressed as:

$$\boldsymbol{x}_{i+1} = \mathcal{F}_{euler}(\boldsymbol{x}_i, t_i, t_{i+1}) = \boldsymbol{x}_i + h_\theta(\boldsymbol{x}_i, t_i) * (t_{i+1} - t_i) \tag{4}$$

However, DDIM [32] quickly became a popular choice of $\mathcal{F}$ for its improved efficiency. Other recent works have tried to rely on approximations or leverage various ideas from the numerical methods literature to design solvers $\mathcal{F}$ that require fewer denoising steps. For instance, Diffusion Exponential Integrator Sampler (DEIS, [42]), and DPM-Solver [19] exploit the special structure of the probability flow ODE to design special solvers where the linear part of the ODE is solved analytically and the non-linear part is solved by incorporating ideas from exponential integrators in the numerical methods literature. Karras et al. [13] leverage the Heun's second order method and demonstrate a favorable tradeoff between number of model evaluations and quality of generated samples for a small number of denoising steps. In this work, SRDS presents an orthogonal improvement to these methods via parallelization, and by default we will assume all our solvers to be DDIM.

## 3 Self-Refining Diffusion Samplers

Attempts to reduce the number of steps in diffusion samplers can provide speedups in sample generation [29, 23], but unfortunately often lead to lower-sample quality. While low-frequency components (in the Fourier sense) of the images may be well-established, the generations miss the high-frequency details that leads to good generations [40]. To fix sample quality while maintaining the latency benefits of reducing the number of steps, we turn to numerical methods introduced in the parallel-in-time integration literature where dynamics with different components having different rates

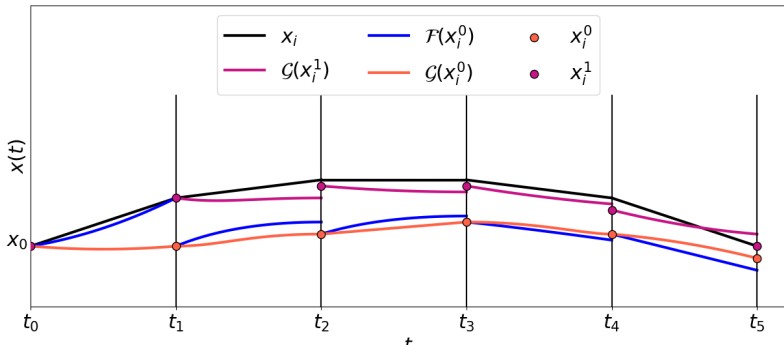

Figure 2: First iteration of the parareal algorithm to solve an example ODE. The black curve represents the desired solution from the fine solver. The magenta dots indicate the running solution after one iteration of predictor-corrector updates. Figure inspiration from Pentland et al. [25].

of convergence has been extensively studied. Specifically, multiple-shooting and multigrid methods have seen success in a wide range of domains from convection-diffusion equations to eigenvalue problems [2, 22, 24, 4] by creating a rough but efficient solve of the prescribed differential equation that can then be iteratively updated via a highly parallelizable simulation. One such algorithm – Parareal [18] – serves as the backbone for our Self-Refining Diffusion Samplers that we describe below.

### 3.1 Parareal Algorithm

Parareal makes use of two solvers: solver $\mathcal{F}$ (called the 'fine solver') provides accurate solutions but is slow to evaluate, and $\mathcal{G}$ (called the 'coarse solver') provides rough solutions but is much quicker.

Parareal targets general purpose initial value problems of forms similar to Equation 1. Consider a partition $(t_0, t_1, ..., t_N = T)$ of the time axis $[t_0, T]$ into N intervals of equal width. Using the same solver notation as above, the goal is to approximate the solution $x_N$ to the initial value problem that would be produced using a sequence of fine solves:

$$x_{i+1} = \mathcal{F}\left(x_i, t_i, t_{i+1}\right), \forall i \in [0, N-1]$$

The key insight of parareal is that we can first use the coarse solver $\mathcal{G}$ to quickly produce a rough trajectory, and this rough solution can be iteratively refined using *parallel* calls to the fine solver $\mathcal{F}$.

Formally, the parareal algorithm begins with a rough estimate of the trajectory, initialzied via a series of coarse solves from $\mathcal{G}$.

$$x_0^0 = x_0 \qquad x_{i+1}^0 = \mathcal{G}\left(x_i^0, t_i, t_{i+1}\right) \forall i \in [0, N-1] \tag{5}$$

where the notation $x_i^0$ denotes the initial estimate of the trajectory from the coarse solver (orange curve in Figure 2).

Parareal then proceeds in iterations until convergence, where each iteration corresponds to a refinement of the trajectory. At each iteration, we solve the differential equation in each of the $N$ time intervals at a higher resolution using the fine solver $\mathcal{F}$, where the initial value for each interval is given by the estimate of the trajectory from the previous iteration. Crucially, these fine solves (blue in Figure 2) can be performed *in parallel*. Lastly, at the end of each iteration, we perform another coarse sequential solve through the trajectory (magenta in Figure 2) and incorporate the results of the fine solves into the running solution for the trajectory using a *predictor-corrector* method, where the coarse solver 'predictions' are 'corrected' via the updates from the parallel fine solves. Formally,

$$x_{i+1}^{p+1} = \mathcal{F}\left(x_i^p, t_i, t_{i+1}\right) + \left(\mathcal{G}\left(x_i^{p+1}, t_i, t_{i+1}\right) - \mathcal{G}\left(x_i^p, t_i, t_{i+1}\right)\right), \quad i = 0, \ldots, N-1 \tag{6}$$

where the notation $x_i^p$ denotes the running estimate of the trajectory at Parareal iteration number $p$.

### 3.2 Self-Refining Diffusion Samplers

Now, we turn our attention back to drawing a sample from our diffusion model, which as discussed corresponds to estimating a solution to the initial value problem as defined in Equation 1.

**Algorithm 1** SRDS: Self-Refining Diffusion Sampler
___
**Require:** Diffusion model $p_\theta$ with denoising steps $N$, tolerance $\tau$, and corresponding DDIM solver $h(\boldsymbol{x}, t_{start}, t_{end}, steps)$

**Ensure:** A sample from $p_\theta$
 1: $\boldsymbol{x}_0^0 \sim \mathcal{N}(\boldsymbol{0}, \boldsymbol{I})$      // Sample initial condition for Initial Value Problem from prior
 2: **for** $i \leftarrow 1$ to $\sqrt{N}$ **do**
 3:     $\boldsymbol{prev}_i \leftarrow h\left(\boldsymbol{x}_{i-1}^0, t_{i-1}, t_i, 1\right)$
 4:     $\boldsymbol{x}_i^0 \leftarrow \boldsymbol{prev}_i$                // Initialize $\boldsymbol{x}$ with a coarse solve
 5: $p \leftarrow 1$                        // SRDS refinement iteration number
 6: **while** $p \leq \sqrt{N}$ **do**
 7:     **for** $i \leftarrow 1$ to $\sqrt{N}$ in parallel **do**
 8:         $\boldsymbol{y}_i \leftarrow h\left(\boldsymbol{x}_{i-1}^{p-1}, t_{i-1}, t_i, \sqrt{N}\right)$            // Perform fine solves in parallel
 9:     **for** $i \leftarrow 1$ to $\sqrt{N}$ **do**                    // Perform a coarse sweep
10:         $\boldsymbol{cur}_i \leftarrow h\left(\boldsymbol{x}_{i-1}^p, t_{i-1}, t_i, 1\right)$
11:         $\boldsymbol{x}_i^p \leftarrow \boldsymbol{y}_i + \boldsymbol{cur}_i - \boldsymbol{prev}_i$            // Take predictor-corrector step
12:         $\boldsymbol{prev}_i \leftarrow \boldsymbol{cur}_i$
13:         **if** $|\boldsymbol{x}_{\sqrt{N}}^p - \boldsymbol{x}_{\sqrt{N}}^{p-1}| < \tau$ **then**                // Check for convergence
14:             break
15:     $p \leftarrow p + 1$
16: **return** $\boldsymbol{x}_{\sqrt{N}}^{p-1}$
___

We leverage the Parareal algorithm in order to parallelize sampling. Our idea is to compute a solution on the $N$-discretization of the interval by instead considering a coarser $\sqrt{N}$-discretization[2] of the interval $[0, T]$. Let $\Delta T = T/\sqrt{N}$, $t_{i+1} = t_i + \Delta T$, partitioning $[0, T]$ into $\sqrt{N}$ intervals. We pick $\sqrt{N}$-step DDIM solver [3] as our fine solver $\mathcal{F}$. In other words, $\mathcal{F}(\boldsymbol{x}_i, t_i, t_{i+1})$ is the result of a $\sqrt{N}$-step DDIM solve propagating $\boldsymbol{x}$ from $t = t_i$ with initial value $\boldsymbol{x}_i$ to $t = t_{i+1}$. We pick 1-step DDIM solver as our coarse solver $\mathcal{G}$. That is, $\mathcal{G}(\boldsymbol{x}_i, t_i, t_{i+1})$ denotes the result of the corresponding 1-step DDIM solve propagating $\boldsymbol{x}$ from $t = t_i$ with initial value $\boldsymbol{x}_i$ to $t = t_{i+1}$ ("step" refers to denoising step involving an $h_\theta$ evaluation).

The SRDS algorithm proceeds as follows. We start with the coarse-solve to generate a rough estimate of the trajectory and final sample, achieved by taking $\sqrt{N}$ DDIM steps in total. Then, each of the $\sqrt{N}$ coarse predictions in the trajectory is simulated at higher resolution with further $\sqrt{N}$ DDIM-iterations *in parallel*, each with an effective time step corresponding to the original $N$-step discretization of the diffusion model. Iterative updates to the running solution then proceed in a manner equivalent to Parareal updates until convergence, as measured by the change in the outputted generation. Our SRDS algorithm is summarized in Algorithm 1.

### 3.3 Convergence Guarantee

The ideal result for diffusion sampling is to get the solution arising from $N$ sequential denoising score steps. SRDS however only starts with a rough solve of the diffusion trajectory taking $\sqrt{N}$ sequential denoising steps. Nevertheless, we can show that each iteration of SRDS (line 6 of Algorithm 1) refines the generated sample and leads us closer to the ideal solution.

**Proposition 1.** The sample output by SRDS converges to the output of the $N$-step sequential solver in at most $\sqrt{N}$ refinement iterations.

A key property of our algorithm is that after $i$ iterations (refinements to the diffusion trajectory) of SRDS, the first $i$ steps of the running trajectory exactly matches the trajectory generated by the sequential solver for the corresponding intervals. Consequently, the algorithm is guaranteed to converge in at most $\sqrt{N}$ iterations. We defer the formal proof to Appendix A. It is also worth

___
[2]It is not required for $N$ to be a perfect square. We can extend the described techniques in a straightforward manner to perform a $\lceil\sqrt{N}\rceil$-discretization instead, with the last interval in the partition having a smaller size. The special choice of $\sqrt{N}$ as the discretization level is further explained in Appendix B.

[3]This refers to a solver that takes $\sqrt{N}$ DDIM steps (or model evaluations) to solve the initial value problem.

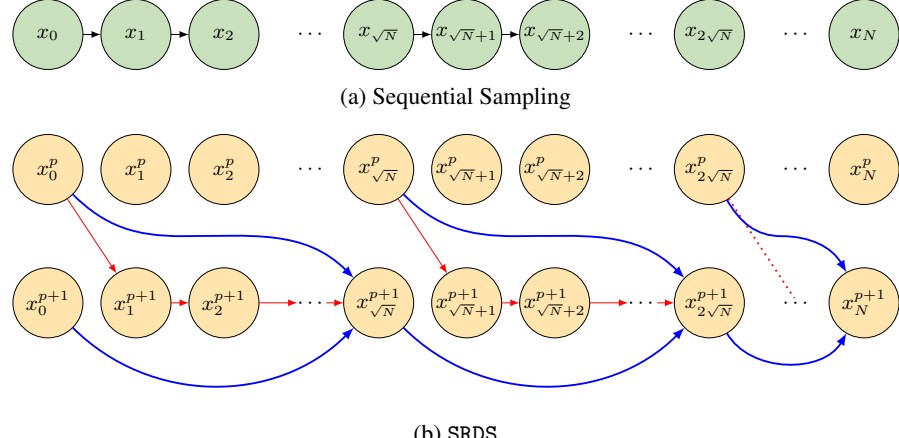

(a) Sequential Sampling

(b) SRDS

Figure 3: Computation graph for diffusion sampling. For SRDS, the red arrows correspond to fine solves, and each block — $[0, \sqrt{N}], [\sqrt{N}, 2\sqrt{N}]$, and so on — can be perform the fine solves independently in parallel. The blue arrows correspond to 1-step coarse solves.

noting that this worst case guarantee is similar in spirit to Proposition 1 in [30] and its generalization (Theorem 3.6 in [37]).

It is worth noting, however, that in practice we observe that the number of refinement iterations required till convergence — which is defined as difference in consecutive sample generations not exceeding a chosen threshold — is *much less* than the worst case bound of $\sqrt{N}$. This early convergence is critical to speedups from SRDS. The exact choice of threshold $\tau$ in line 13 of Algorithm 1, is a hyperparameter that is empirically chosen so as to avoid measurable degradation in sample quality.

### 3.4 Batched Inference and Pipelining

SRDS benefits from two key features to reduce latencies: batched inference and pipelining.

First, the fine solves that are used in order to refine the trajectories implementation-equivalent DDIM-steps, which means that they can be performed in a batched manner even for a single sample generation. This parallelization allows for a single sample generation to incur the benefits of batched inference, introducing higher device utilization or device parallelism.

Secondly, we observe that the dependency graph for SRDS enables pipelined parallelism. As outlined in Figure 3, we find that $\mathcal{F}\left(\boldsymbol{x}_i^p, t_i, t_{i+1}\right)$ and $\mathcal{G}\left(\boldsymbol{x}_i^p, t_i, t_{i+1}\right)$ both only depend on $\boldsymbol{x}_i^p$. The tasks for computing $\mathcal{F}\left(\boldsymbol{x}_j^p, t_i, t_{i+1}\right)$ and $\mathcal{G}\left(\boldsymbol{x}_i^p, t_i, t_{i+1}\right)$ can be spawned as soon as $\boldsymbol{x}_j^i$ is computed, without waiting for the entire predictor-corrector mechanism to finish updating the SRDS solution for iteration $i$. This leads to an efficiently pipelined version of the algorithm, further speeding up the sampling process by a factor of two. See Figure 4 for an illustration of this pipelined algorithm with $N = 16$. Pipelining furthers the benefits of batched inference as the coarse solver is simply a DDIM-step with a larger time-step, so it can be batched with fine solves when applicable.

### 3.5 Sampling Latency

**Proposition 2. [Worst-Case Behavior]** Ignoring GPU overhead, the worst case wall-clock time of generating a sample through SRDS is no worse than that of generating through sequential sampling.

Referring to the pipelined implementation of SRDS, it is easy to see that the fine solve $\mathcal{F}\left(\boldsymbol{x}_i^i, t_i, t_{i+1}\right)$ starts immediately after $\mathcal{F}\left(\boldsymbol{x}_i^{i-1}, t_{i-1}, t_i\right)$. Subsequently, from Proposition 1, it then follows that in the worst case, the final sample of SRDS $\boldsymbol{x}_{\sqrt{N}}^{\sqrt{N}}$ is computed at time $\sqrt{N} \cdot \sqrt{N} = N$ as desired. A formal argument can be found in Appendix A. It is worth noting, however, that this property of SRDS comes at the cost of much higher parallel compute compared to sequential sampling.

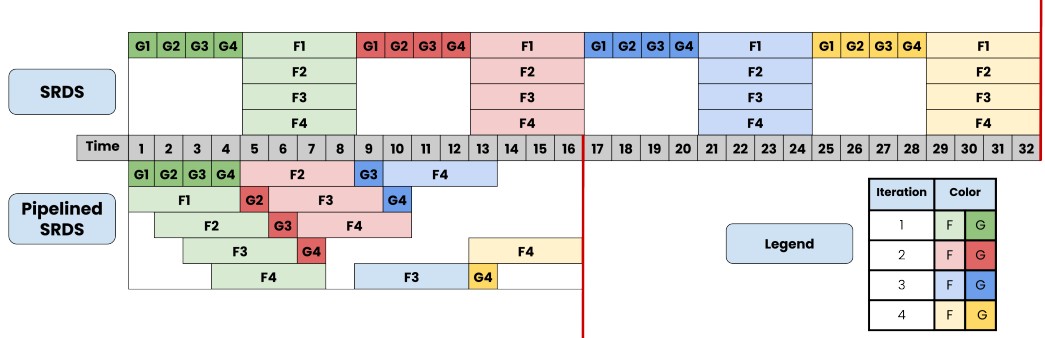

Figure 4: Illustration of the pipelined version of the SRDS algorithm on $N = 16$ denoising steps, which results in a direct 2x speedup compared to the vanilla version.

## 3.6 Memory and Communication Overhead

**Proposition 3. [Memory]** SRDS requires memory corresponding to $\mathcal{O}(\sqrt{N})$ model evaluations.

Once again referring to the pipelined implementation of SRDS, it is easy to see that at any given time there is at most one model evaluation corresponding to a coarse solve, and up to $\sqrt{N}$ parallel model evaluations corresponding to the fine solves. It is worth contrasting this with the quadratically higher $\mathcal{O}(N)$ memory requirement of the full ParaDiGMs algorithm in [30], necessitating the use of sliding window tricks to reverse the process in a piece-wise fashion.

It is finally worth noting that there is minimal inter-GPU communication in SRDS. In particular, at most one sample is passed between adjacent GPUs in each SRDS iteration. Once again, it is worth contrasting this with ParaDiGMs algorithm, which – by its use of parallel prefix sum operations to sync the solutions at each Picard iteration – incurs greater GPU communication overhead. See Appendix D for more discussion.

## 4 Experiments on Diffusion Image Generation

To showcase the capabilities of the prescribed SRDS algorithm, we apply the diffusion sampler to pretrained diffusion models and present the difference in sample time and quality to ensure that applied convergence criteria do not reduce generation metrics. We start with pixel-based diffusion before expanding experiments applied to latent methods such as StableDiffusion-v2. Across the range of tasks, we show consistent speedups while maintaining quality of sample generation.

In this section, we perform an extensive comparison with ParaDiGMs [30] as our baseline. Nonetheless, we provide a high level empirical comparison to our concurrent work ParaTAA[37] in Appendix E, where we demonstrate the superiority of SRDS.

### 4.1 Pixel Diffusion - Image Generation

We start with pixel-space diffusion models. In particular, we test our SRDS algorithm and demonstrate capabilities in performing diffusion directly on the pixel space of 128x128 LSUN Church and Bedroom [41], 64x64 Imagenet [5], and 32x32 CIFAR [16] using pretrained diffusion models [29], which all use $N = 1024$ length diffusion trajectories.

We measure the convergence via $l_1$ norm in pixel space with values [0, 255]. We conservatively set $\tau = 0.1$, meaning that convergence occurs when on average each pixel in the generation differs by only 0.1 after a refinement step (see Appendix F for an ablation on choice of $\tau$). Through our experiments, we quantitatively showcase how the SRDS algorithm can provide signficant speedups in generation without degrading model quality (as measured by FID score [9] on 5000 samples). As seen in Table 1, SRDS remarkably converges in 4-6 iterations across all datasets; this corresponds to roughly $150 - 200$ effective serial steps (counting all model evaluations simultaneously performed in parallel as one evaluation), which is only $15 - 20\%$ of the serial steps required by a sequential solve

Table 1: Evaluating FID score (lower is better) of SRDS on various datasets using 5000 samples generated using a DDIM solver. Effective serial evals refers to the number of serial model evaluations taken by the pipelined SRDS algorithm (counting all model evaluations simultaneously performed in parallel as one evaluation) . Total evals refers to the total number of model evaluations.

| | Sequential | | SRDS | | | |
| Dataset | Serial Evals | FID Score | SRDS Iters | Eff. Serial Evals | Total Evals | FID Score |
|---|---|---|---|---|---|---|
| LSUN Church | 1024 | 12.8 | 5.7 | 209 | 5603 | 12.8 |
| LSUN Bedroom | 1024 | 10.0 | 5.8 | 212 | 5692 | 10.0 |
| Imagenet | 1024 | 9.0 | 4.6 | 175 | 4612 | 9.0 |
| CIFAR | 1024 | 7.6 | 3.7 | 147 | 3771 | 7.6 |

Table 2: CLIP scores of SRDS on StableDiffusion-v2 over 1000 samples from the COCO2017 captions dataset, with classifier guidance $w = 7.5$, evaluated on ViT-g-14. Time is measured on 4 A100 GPUs **without pipeline parallelism**, showcasing speedups with early convergence of the SRDS sample.

| | Sequential | | | Vanilla SRDS | | | | | |
| Stable Diffusion-v2 | Serial Evals | CLIP Score | Time per Sample | Max Iter | Eff. Serial Evals | Total Evals | CLIP Score | Time per Sample | Speedup |
|---|---|---|---|---|---|---|---|---|---|
| DDIM | 100 | 31.9 | 4.6 | 1 | 19 | 119 | 31.9 | 2.0 | 2.3x |
| DDIM | 25 | 31.7 | 1.2 | 1 | 9 | 34 | 31.4 | 0.8 | 1.5x |
| DDIM | 25 | 31.7 | 1.2 | 3 | 17 | 74 | 31.9 | 1.7 | 0.7x |

($N = 1024$). We clarify that effective serial evaluations is referred to as *Parallel Iters* in [30] and *Steps* in [37].

While we are pretty conservative above in measuring convergence through distance in pixel space, we can also simply limit the number of SRDS iterations to $1 - 2$ and achieve further speedups without measurable degradation in sample quality. See Appendix F for more details.

It is once again worth noting that this improved latency from parallelization comes at the cost of greater number of total model evaluations compared to a regular sequential solver. However, this tradeoff enables the diffusion models for many other use cases such as real-time image or music editing and trajectory planning in robotics. Moreover, we often empirically observe that SRDS provides reasonable predictions within a single Parareal iteration; here, the total number of model evaluations is only slightly larger than the serial approach (increasing from $n$ to $n + 2\sqrt{n}$). Lastly, it is also worth noting that many users are often agnostic to inference time GPU compute costs as they are orders of magnitude lower than training compute costs anyway.

## 4.2 Latent Diffusion - Image Generation

Finally, we turn to latent diffusion models, in particular StableDiffusion-v2 [27], where evaluations of the CLIP score over 1000 random samples show how SRDS maintains sample quality while improving the number of parallel iterations required per sample, with summary metrics presented in Table 2. As the SRDS algorithm has small GPU overhead, we achieve measured wallclock time improvements with a Diffusers compatible implementation [38]. It is worth nothing that while we focus on DDIM here (as in the rest of the writing), we show speedups by readily incorporating other solvers into SRDS in Appendix C.

For the test bed of latent diffusion models, we explore the convergence properties of our SRDS algorithm, with the average CLIP score plotted against the number of iterations in Figure 5. For shorter sequences of length 25 (left), the corresponding SRDS sampler converges after approximately 3 iterations. However, for longer sequences of length 100 (right) the sampler has converged after a single SRDS iteration, showcasing the capabilities of our algorithm improves with longer trajectories.

Next, we demonstrate the additional speedup that pipeline parallelism can bring to SRDS. We implement a slightly suboptimal version of pipelined SRDS for StableDiffusion and already observe

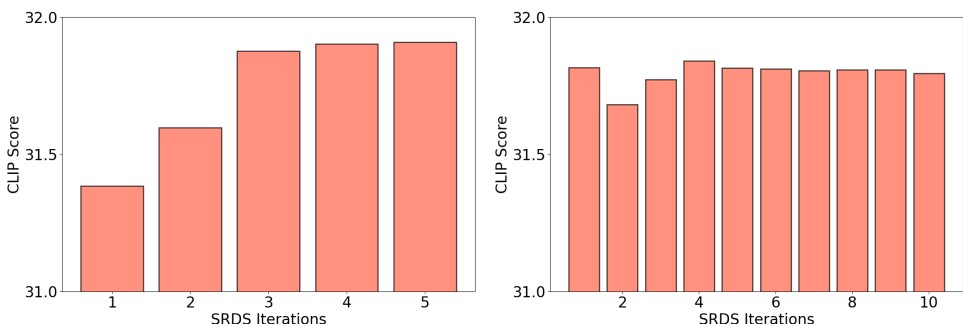

Figure 5: Convergence of the SRDS algorithm for a trajectory of length 25 (left) and 100 (right) showcase how early termination of the algorithm can yield equivalent sample quality. In particular, longer trajectories with increased parallelism appear to converge faster.

Table 3: Evaluation of additional speedup offered by pipelined version of SRDS.

| Method | Serial Model Evals | SRDS | | Pipelined SRDS | |
| | | Eff. Serial Evals | Time Per Sample | Eff. Serial Evals | Time Per Sample |
| --- | --- | --- | --- | --- | --- |
| DDIM | 961 | 93 | 12.30 | 63 | 10.31 |
| DDIM | 196 | 42 | 3.30 | 27 | 2.85 |
| DDIM | 25 | 15 | 0.82 | 9 | 0.69 |

Table 4: Comparison of wallclock speedups offered by Pipelined SRDS and ParaDiGMS with various thresholds, with respect to Serial image generation. These StableDiffusion experiments are performed on identical machines (4 40GB A100 GPUs) for a fair comparison.

| Method | Serial | | Pipelined SRDS | ParaDiGMS | | |
| | Model Evals | Time Per Sample | Time Per Sample | Threshold 1e-3 | Threshold 1e-2 | Threshold 1e-1 |
| --- | --- | --- | --- | --- | --- | --- |
| DDIM | 961 | 44.88 | 10.31 (4.3x) | 275.29 | 20.48 | 14.30 |
| DDIM | 196 | 9.17 | 2.85 (3.2x) | 29.45 | 5.08 | 3.42 |
| DDIM | 25 | 1.18 | 0.69 (1.7x) | 1.98 | 1.51 | 0.77 |

significant speedups as seen in Table 3; with some more engineering effort[4], we can further push towards extracting the full potential of pipelining. However, as this already beats the baselines, this sufficiently demonstrates the benefits of SRDS[5].

Furthermore, for our main baseline ParaDiGMs, we also perform more extensive evaluation to evaluate both methods on equal hardware to more clearly demonstrate the benefits of SRDS. In Table 4, we demonstrate that SRDS consistently beats ParaDiGMS on wallclock speedups. Though the authors of [30] uses a convergence threshold of $1e-3$, we show that SRDS can provide impressive speedups even when compared to significantly relaxed ParaDiGMS thresholds of $1e-1$.

Finally, we finally provide a few sample generations on standard text prompts from DrawBench [28] in Figure 6 and additional examples of convergence similar to Figure 1 in the Appendix G.

## 5 Related Work

Recent literature on diffusion models has focused heavily on reducing the cost of sampling. Techniques such as higher order methods [13] and exponential integrators [42] have been proposed as

---

[4]The suboptimality of the implementation arises from the use of a single device to coordinate the pipeline parallelism and device transfers (arising as an artifact from torch.multiprocessing). A more complete implementation would instead use ring-like communication between devices rather than wait on the coordinator.

[5]We note that the number of denoising steps in the experiments is chosen to be perfect squares merely for convenience. SRDS is general and applies to any number of denoising steps.

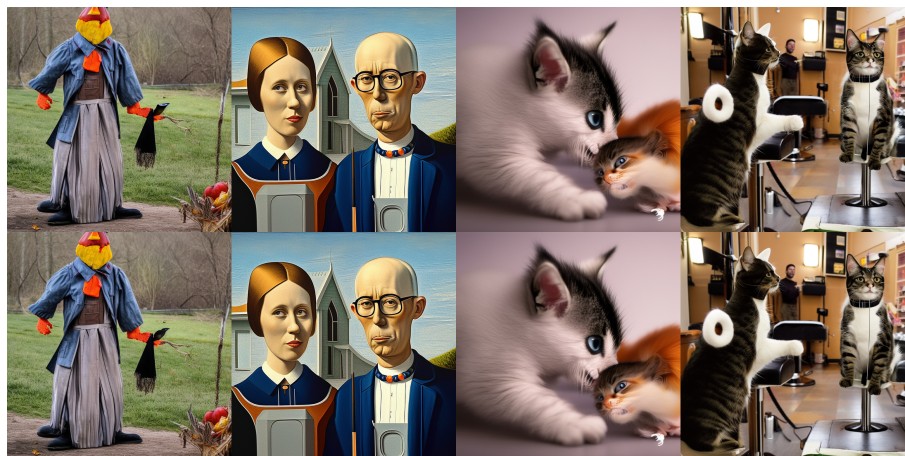

Figure 6: Sample generation from StableDiffusion-v2 with the SRDS algorithm with text prompts based on examples from DrawBench. We plot the early converged SRDS figure (top) and the result of the serial trajectory (bottom); the two rows are essentially indistinguishable, highlighting the approximation-free nature of SRDS.

strategies for reducing the model evaluations required in order to build high-quality samples without any additional training. When additional training is possible, other works have proposed distillation [29], quantization [17], and consistency [34] as alternate objectives to further speed up sample generation. For the purposes of this paper, we view these approaches as orthogonal, as the resultant models could be simulated with SRDS for potential benefits from combining methods.

As discussed throughout the paper, this work is most cloesly related to the ParaDiGMS sampler method developed by Shih et al. [30] for parallel sampling of diffusion models. The two works take a similar approach by building off popular parallel-in-time integration methods in order to achieve lower latencies in simulation. In particular, ParaDiGMS builds on Picard iterations to converge on trajectories; we, however, build on Parareal method that performs multiresolution along the time axis for faster sampling. Parareal has been well-explored [18, 25, 20, 7] though with limited theoretical guarantees only spanning certain cases such as the heat equation and Navier-Stokes equation [8, 36]; our work is the first to apply this algorithm to diffusion models.

## 6   Conclusion

**Limitations:**   Similar to previous iterations of parallel-in-time integration methods, SRDS make use of additional compute that can be used in parallel in exchange for faster latencies of sampling. That is to say, the total number of model evaluations in comparison to standard diffusion modelling increases in exchange for lower latencies. The additional compute may be reasonable in applications such as small-batch sampling where the additional cost can be hidden through better device utilization (e.g. sampling of a single image or trajectory in robotics). Alternatively, the responsiveness of real-time image editing may make parallel sampling an appealing option for cost-insensitive users.

**Future Directions:**   This work opens up a ton of interesting open questions for future exploration. Firstly, while fast convergence of parareal-style algorithms has only been proven for very limited settings, it will be extremely interesting to derive convergence guarantees specifically for the diffusion process. This has the potential to further our understanding of the nature of the ODE/SDE that governs the reverse process. Another natural direction is to explore the effects of employing higher levels of discretization and other multigrid methods such as $F$-cycles and $W$-cycles. As alluded to in Section 3.2, one could not only further study the optimal choice of second level of discretization, but also consider novel schedules that involve partitioning the diffusion trajectory into intervals of varying sizes. Lastly, it is worth highlighting that by serving a highly modular and interoperable framework, SRDS unlocks a vast landscape of interesting coarse/fine solver combinations. For instance, one can use a DDIM solver to perform the parallel refinement steps, while using a progressively distilled model [29] or consistency trajectory model [15] as the coarse solver in SRDS.

## Acknowledgments and Disclosure of Funding

We thank the reviewers for their thoughtful feedback towards improving this paper. We also thank Aryaman Arora, Harshit Joshi, Ken Liu, Rajeshwari Jadhav, Rohit Nema, and Yanzhe Zhang for their helpful discussions and support during various stages of the project. This project was funded in part by ARO (W911NF-21-1-0125), ONR (N00014-23-1-2159), and the CZ Biohub.

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

# A  Proofs

**Proposition 1.  [Convergence Guarantee]**  The sample from SRDS converges to the output of the slow sequential solver in at most $\sqrt{N}$ refinement iterations.

*Proof.* We show, by induction, that $\boldsymbol{x}_i^p$ converges in $i$ iterations of SRDS for all $i \in [0, N-1]$. Further, $\boldsymbol{x}_i^p = \mathcal{F}\left(\boldsymbol{x}_{i-1}^{p-1}, t_{i-1}, t_i\right)$ for all $p \geq i$, implying that the final sample indeed corresponds to the desired sample from $\mathcal{F}$. The base case of $i = 0$ follows trivially from the initialization (initial condition). To prove the second base case of $i = 1$, notice that $\boldsymbol{x}_0^p = \boldsymbol{x}_0$ for all $p$, implying that $\mathcal{G}\left(\boldsymbol{x}_0^{p-1}, t_0, t_1\right)$ is constant for all $p \geq 1$. Consequently,

$$\boldsymbol{x}_1^p = \mathcal{F}\left(\boldsymbol{x}_0^{p-1}, t_0, t_1\right) + \left(\mathcal{G}\left(\boldsymbol{x}_0^p, t_0, t_1\right) - \mathcal{G}\left(\boldsymbol{x}_0^{p-1}, t_0, t_1\right)\right), \quad \forall p \geq 1$$
$$= \mathcal{F}\left(\boldsymbol{x}_0, t_0, t_1\right), \quad \forall p \geq 1$$

as desired.

Assume by the induction hypothesis that for some fixed $i$, $\boldsymbol{x}_i^p = \mathcal{F}\left(\boldsymbol{x}_{i-1}^{i-1}, t_{i-1}, t_i\right), \forall p \geq i$. Then, $\forall p \geq i$, we have that

$$\boldsymbol{x}_{i+1}^{p+1} = \mathcal{F}\left(\boldsymbol{x}_i^p, t_i, t_{i+1}\right) + \left(\mathcal{G}\left(\boldsymbol{x}_i^{p+1}, t_i, t_{i+1}\right) - \mathcal{G}\left(\boldsymbol{x}_i^p, t_i, t_{i+1}\right)\right)$$
$$= \mathcal{F}\left(\boldsymbol{x}_i^i, t_i, t_{i+1}\right) + \left(\mathcal{G}\left(\mathcal{F}\left(\boldsymbol{x}_{i-1}^{i-1}, t_{i-1}, t_i\right), t_i, t_{i+1}\right) - \mathcal{G}\left(\mathcal{F}\left(\boldsymbol{x}_{i-1}^{i-1}, t_{i-1}, t_i\right), t_i, t_{i+1}\right)\right)$$
$$= \mathcal{F}\left(\boldsymbol{x}_i^i, t_i, t_{i+1}\right)$$

as desired. $\qquad\square$

**Proposition 2.  [Worst-Case Sampling Latency]**  Ignoring GPU overhead, the worst case wall-clock time of generating a single sample through SRDS is no worse than that of generating a single sample through sequential sampling.

*Proof.* Consider the unit of time to be the time taken for one denoising step (or one model evaluation). Referring to the pipelined implementation of SRDS, it is easy to see via a straightforward inductive argument that the $\sqrt{N}$-step fine solve $\mathcal{F}\left(\boldsymbol{x}_{\sqrt{N}}^p, t_i, t_{i+1}\right)$ ends at time $\frac{N}{\sqrt{N}}p + \sqrt{N} - p$. From Proposition 1, it then follows that in the worst case, the final sample of SRDS $\boldsymbol{x}_{\sqrt{N}}^{\sqrt{N}}$ is computed at time $\frac{N}{\sqrt{N}}\sqrt{N} + \sqrt{N} - \sqrt{N} = N$ as desired. $\qquad\square$

**Proposition 3.  [Memory]**  SRDS requires memory corresponding to $\mathcal{O}(\sqrt{N})$ denoising model evaluations.

*Proof.* In the pipelined implementation of SRDS, it is easy to see that at any given timestep there is at most one model evaluation corresponding to a coarse solve. Further, the number of parallel model evaluations corresponding to the fine solves is upper bounded by the coarse discretization (or the number of "blocks"), which is $\sqrt{N}$. Thus, the memory used by SRDS corresponds to at most $\sqrt{N} + 1 = \mathcal{O}(\sqrt{N})$ model evaluations. $\qquad\square$

# B Choice of Coarse Resolution

The choice of resolution for the coarse solve is not arbitrary. For practical implementations, since we use the same denoiser (say, DDIM) for both the coarse and fine solves, we choose $\sqrt{N}$ as an optimal choice in the runtime sense (assuming constant number of iterations till convergence[6]). At a high level, this choice stems from the fact that we want to balance out the time the it takes to run all the fine solves in parallel and the time it takes perform one set of sequential predictor-corrector steps through the trajectory.

**Proposition 4.** **[Optimal Coarse Resolution]** The speed of an `SRDS` iteration is maximized for $B \approx \sqrt{N}$.

*Proof.* Let $k$ denote the number of `SRDS` iterations until convergence, let $\tau$ denote the cost of one denoising step or model evaluation, and let $1 < B < N$ denote the "block-size": that is, the second scale of discretization. For the 1-step coarse solve, each `SRDS` iteration incurs a runtime cost of $1 \cdot \lceil \frac{N}{B} \rceil \cdot \tau$. For the $B$-step fine solves, as each of the $\lceil \frac{N}{B} \rceil$ fine solves are independently executed in parallel, each `SRDS` iteration incurs a runtime cost of $B \cdot 1 \cdot \tau$. The baseline runtime for sequentially sampling from the diffusion model is $N \cdot \tau$. Thus, the runtime speedup (ignoring parallelization overhead) is $\frac{N \cdot \tau}{k\left(\lceil \frac{N}{B} \rceil \cdot \tau + B \cdot \tau\right)} = \frac{N}{k\left(\lceil \frac{N}{B} \rceil + B\right)}$. For a fixed value of $k$, it is easy to see that this quantity is concave in $B$ and is maximized by choosing $B \approx \sqrt{N}$. $\square$

It is worth noting, however, that if we use solvers of different latencies for the coarse and fine steps, a modifed analysis is required to incorporate differences in denoising step times for the two solvers. Consequently, $\sqrt{N}$ might no longer be the optimal choice of coarse resolution.

# C Incorporation of other Solvers

It is worth emphasizing again that SRDS provides an orthogonal improvement when compared to the other lines of research on accelerating diffusion model sampling. In particular, while the main experiments (and writing) were focused on DDIM, `SRDS` is compatible with the other solvers and they can be readily incorporated into `SRDS` to speed up diffusion sampling. For example, below we show that SRDS is directly compatible with other solvers such as DDPM (often requiring more steps than DDIM) and DPMSolver (often requiring fewer steps than DDIM) and can efficiently accelerate sampling in both cases. We demonstrate this on StableDiffusion in Table 5. We also highlight that the Diffuser-compatible implementation requires only minor modification to the arguments of the solver, suggesting that `SRDS` will also be easy to extend out-of-the-box to other methods that the community develops.

Table 5: Evaluation of `SRDS` with various off-the-shelf solvers.

| Model | Sequential | | SRDS | | Speedup |
| | Model Evals | Time Per Sample | Eff Serial Evals | Time Per Sample | |
| --- | --- | --- | --- | --- | --- |
| DDPM | 961 | 44.68 | 93 | 12.30 | 3.63x |
| DDPM | 196 | 9.03 | 42 | 3.26 | 2.76x |
| DPM Solver | 196 | 10.30 | 42 | 3.49 | 2.95x |
| DPM Solver | 25 | 1.31 | 15 | 0.88 | 1.48x |
| DDIM | 196 | 9.17 | 42 | 3.30 | 2.77x |
| DDIM | 25 | 1.18 | 15 | 0.82 | 1.43x |

---

[6]It is possible that a different choice of $B$ might actually be optimal by enabling better flow of information down the computation graph and thereby resulting in lower number of iterations till convergence $k$. However, in our experiments (as also noted in Section 4), we observe that `SRDS` often converges for small $k \in [2, 4]$, validating our empirical choice.

# D    Memory Utilization

For a $T$ step denoising process, ParaDiGMS needs to perform T model evaluations in parallel with subsequent computations needing information about all previous evaluations, while SRDS only requires $\sqrt{T}$ parallel evaluations (which fits comfortably in GPU memory) and requires much lesser communication between GPUs. While the prohibitively large memory requirement can be combated with a sliding window method, the significantly larger communication overhead remains because at every step of Paradigms, an AllReduce over all devices must be performed in order to calculate updates to the sliding window. (For instance, even when ParaDiGMS reduces Eff. Serial Steps by 20x, the obtained speedup is only 3.4x). This is in contrast to the independent fine-solves in parareal that only need to transfer information for the coarse solve.

Below in Table 6, we demonstrate how the minimal memory and communication overhead of SRDS shines through as we are able to achieve better device utilization as we increase the number of available GPUs. The following experiment was performed on 40GB A100s and used a generous 1e-2 threshold for ParaDiGMS.

Table 6: Evaluating the device utilization of SRDS in comparison to `ParaDiGMS`.

|  | Devices | Serial Model Evals | SRDS | | ParaDiGMS | |
|---|---|---|---|---|---|---|
|  |  |  | Eff Serial Evals | Time Per Sample | Eff Serial Evals | Time Per Sample |
| DDIM | 1 | 25 | 15 | 1.62 | 16 | 2.71 |
| DDIM | 2 | 25 | 15 | 1.08 | 16 | 2.01 |
| DDIM | 4 | 25 | 15 | 0.82 | 16 | 1.51 |

# E    Comparison to `ParaTAA`

We demonstrate the superiority of SRDS to baselines `ParaDiGMS` [30] and `ParaTAA` [37]. Here, we demonstrate the high-level superiority of SRDS solely by using the results published by the authors in [30] (Table 5) and [37] (Table 1).

In the table 7 below, we show that SRDS offers better wall-clock speedups (over sequential) in sample generation time for StableDiffusion when compared to [30] and [37]. We clarify that the reported speedup for each method is with respect to sequential solve on the same machine that the corresponding parallel method was evaluated. Our results are particularly impressive given that the authors of [30] used 8x 80GB A100s for the evaluation and the authors of [37] used 8x 80GB A800 for the same, while we (SRDS) only used 4x 40GB A100 for the evaluation due to computational constraints. (For interpretation purposes, recall that a sequential solve is not compute/memory bound and doesn't benefit significantly from additional GPU compute, whereas the parallel methods certainly do!) We would also like to highlight the superiority of SRDS over the baselines in the regime of small number of denoising steps (25) as being particularly impactful.

Table 7: Speedup in wallclock time for single sample generation offered by SRDS compared to `ParaDiGMS` and `ParaTAA`

| Denoising Steps | ParaDiGMS | ParaTAA | Pipelined SRDS |
|---|---|---|---|
| DDIM - 100 | 2.5x | 1.92x | 2.73x |
| DDIM - 25 | 1.0x | 1.17x | 1.72x |

# F    Additional Convergence Experiments

In this section, we further evaluate the convergence properties of SRDS. First, in Table 8, we analyze how the sample quality varies as we vary the tolerance threshold $\tau$.

Table 8: Evaluating the effect of SRDS tolerance parameter $\tau$ on sample quality for a pretrained diffusion model on 128x128 LSUN Church. The metric used is Kernel Inception Distance (KID, lower is better) over 1000 random samples generated using DDIM solvers.

| Method - $\tau$ | SRDS Iterations | Eff. Serial Iters | Model Evals | KID Score |
|---|---|---|---|---|
| Sequential - N/A | N/A | 1024 | 1024 | 0.0146 |
| SRDS - 0.1 | 5.7 | 209 | 5603 | 0.0146 |
| SRDS - 0.5 | 4.3 | 165 | 4334 | 0.0147 |
| SRDS - 1.0 | 3.7 | 147 | 3771 | 0.0146 |

Next, we analyze how the FID score of generated samples varies as a function of the number of SRDS iterations. As shown in Figure 7, we observe rapid convergence of the FID score to the value obtained by sequential sampling (12.8) within a few SRDS iterations.

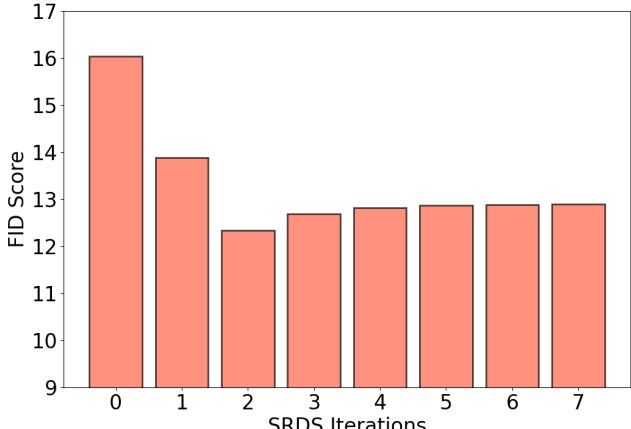

Figure 7: Convergence of the SRDS algorithm on LSUN Church.

# G    Additional Samples from SRDS

Various samples from Drawbench prompts are provided in Figure 8.

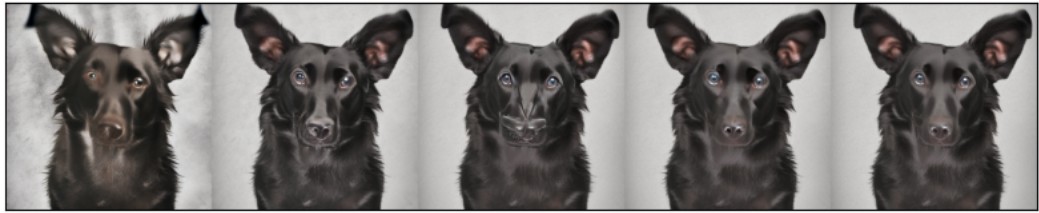

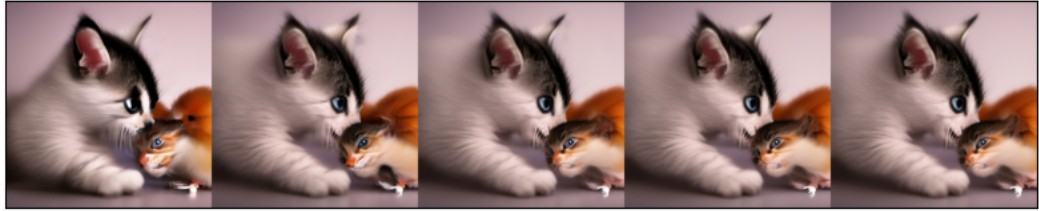

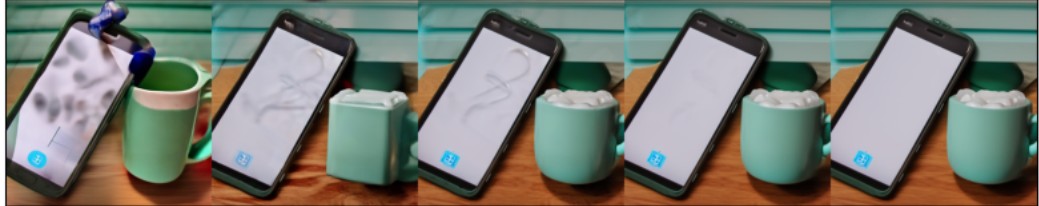

Figure 8: Convergence of the SRDS algorithm on various samples from DrawBench.

