# OpenReview forum: "Self-Refining Diffusion Samplers: Enabling Parallelization via Parareal Iterations"
_NeurIPS.cc/2024/Conference — NeurIPS 2024 poster_

### Official Review · Reviewer_KhBd · 2024-07-09

**Soundness:** 2
**Presentation:** 3
**Contribution:** 2
**Rating:** 6
**Confidence:** 5

**Summary:**

This paper proposes a new method for parallelizing the sampling process of diffusion models, offering a novel way to trade compute for speed. In particular, it relies on the parareal iteration in this setting, where the main idea is to split the solving timesteps into multiple groups and then update them in parallel. To ensure that the output from these timestep groups matches the exact solution, parareal performs predictor-corrector steps after each group is updated. Experiments demonstrate the efficacy of applying parareal iteration in speeding up the sampling process.

**Strengths:**

1. This paper demonstrates a very interesting direction by introducing the parareal iteration into diffusion models.
2. The experiments indeed demonstrate the efficacy of this approach.

**Weaknesses:**

1. I think the author misses a rigorous discussion comparing the parareal algorithm with ParaDiGMS [1] and also completely ignores a more recent reference [2]. It is clear that all these works, including parareal, focus on parallelizing the sampling process via certain forms of fixed-point iteration, although the computing graphs of these algorithms differ. The only discussion in the current manuscript is that ParaDiGMS is highly memory-intensive and must use a sliding window. While this statement is certainly true, I believe it is not a disadvantage of ParaDiGMS and does not imply that ParaDiGMS is inferior to parareal. I do not see why using a sliding window should be considered a disadvantage.

2. As far as I can see, the Eff. Serial Evals in Table 1 and Table 2 should correspond to the Parallel Iters in [1] and Steps in [2], am I correct? Because Eff. Serial Evals essentially represent the number of batch evaluations on the models. If this is the case, based on the current numbers in Table 1, it seems quite inferior to the results in [1] and [2]. Of course, I understand that the batch size may be different in this work, but I still wish the author would provide a detailed discussion on this point.

3. Proposition 1 in the manuscript is somewhat trivial and exists in many existing works. Note that each refinement consists of $\sqrt n$ sequential evaluation, so the parareal will also require at most $\sqrt n \times \sqrt n=n$ sequential evaluation on the models to converge. I think the author should mention that this result is fundamentally equivalent to Proposition 1 in [1], and a more general result is found in Theorem 3.6 in [2]. The key point is that when performing fixed-point iteration for an $n$-order triangular system, it will certainly converge within $n$ steps, and this bound can not be further improved without extra assumption.



[1] Shih, Andy, et al. "Parallel sampling of diffusion models." Advances in Neural Information Processing Systems 36 (2024).

[2] Tang, Zhiwei, et al. "Accelerating parallel sampling of diffusion models." Forty-first International Conference on Machine Learning. 2024.

**Questions:**

1. In my understanding, the technique in Parareal have a different spirit in parallelization with [1][2]. For example, consider that we wish to do n-step sampling with $\sqrt n$ batchsize. Given a initial sequence $[x_1,...,x_n]$, parareal first group them into $g_1=[x_1,...,x_{\sqrt n}],...,g_{\sqrt n}=[x_{n-\sqrt n},x_n]$. The parallelism is that to update $g_1$, ..., $g_n$ in parallel, while the update of each $g_i$ consists of $\sqrt n$ sequential solving. On the other hand, [1][2] consider update $[x_{i},...,x_{i+\sqrt n}]$ in parallel until convergence. It is not clear to me which one is better from the theoretical aspects, as they have the same worst-case convergence bound. I suspect that it should depends on the form of ODE and also the batchsize? I wonder if the authors could provide some thought and discussion on this point?

2. Do you think parareal can be applied to the SDE sampler as [1] and [2] did?


[1] Shih, Andy, et al. "Parallel sampling of diffusion models." Advances in Neural Information Processing Systems 36 (2024).

[2] Tang, Zhiwei, et al. "Accelerating parallel sampling of diffusion models." Forty-first International Conference on Machine Learning. 2024.

**Limitations:**

See my comments on weaknesses and questions above. This work lacks a correct and thorough discussion comparing it to existing works.

---

> ### Author Rebuttal · Authors · 2024-08-07
>
> We thank the reviewer for the review, and are excited to hear that the reviewer finds the novel ideas presented to be very interesting, and that the experiments indeed demonstrate the efficacy of this approach! We respond to the concerns below:
>
> > Baselines
>
> Please refer to the combined response to all reviewers. We thank the reviewer for bringing [2] to our attention; we were not aware of this very recent work (ICML 2024) at the time of writing this paper. However, their method uses completely different techniques (namely, formulating and solving a triangular system in a specialized way) to solve the same problem. We have discussed this work in the common response, and will make sure to include a detailed discussion in the final manuscript.
>
> > As far as I can see, the Eff. Serial Evals in Table 1 and Table 2 should correspond to the Parallel Iters in [1] and Steps in [2] ... based on the current numbers in Table 1, it seems quite inferior to the results in [1] and [2]. Of course, I understand that the batch size may be different in this work, but I still wish the author would provide a detailed discussion on this point.
>
> Thank you for highlighting the difference in notation between the papers. You are correct that the Eff. Serial Evaluations correspond to the Parallel Iters in [1] and Steps in [2], where all refer to the number of batched evaluations. We will be sure to clarify this notation in updates to the paper.
>
> The reasons for the seemingly high number of Eff. Serial Iterations is twofold:
>
> - *Stricter convergence criteria for SRDS*: We note that there is a considerable difference in the how the convergence criteria is calculated for ParaDiGMS and SRDS. ParaDiGMS using a per step tolerance proportional to the scheduler’s noise magnitude and aggressively (and irrecoverably) slides the sliding window if an early step has “converged”; this gives very loose guarantees of the final generated image. On the other hand, SRDS only directly considers differences in the final output representation to decide convergence. This strict/conservative threshold for our pixel diffusion results in Table 1 causes larger number of effective serial iteration. One can comfortably relax this threshold and still maintain good sample quality (as seen in the ablation study in Appendix C). Alternatively, as we show for StableDiffusion, one can instead limit the number of iterations and drastically reduce the Eff. Serial Evaluations (comparable to ParaDiGMS) with no measurable degradation in sample quality! In fact, even for pixel-based diffusion models, we can similarly limit the number of SRDS iterations to 1 or 2 without degradation in sample quality (Figure 7 in Appendix C).
>
> - *Batch size*: As the reviewer hinted correctly, ParaDiGMS has a much higher needed batch size. Specifically, for a T step denoising process, ParaDiGMS needs to perform T model evaluations in parallel, while SRDS only requires $\sqrt{T}$ evaluations, which fits comfortably in GPU memory. To combat this using sliding windows, ParaDiGMS is able to arbitrarily increase the compute and create larger sliding windows (in exchange for reducing the number of serial iterations). It is worth noting that the comparison performed for the number of serial evaluations in the ParaDiGMS paper is on a system of 8 A100 80Gb GPUs whereas our experiments are performed on 4 A100 40Gb GPUs. To shed light on this difference in platforms, we also try to provide a more fair comparison in our experiments in the combined reviewer response, and we will add this clarification to the revised manuscript.
>
>
> > fixed-point iteration
>
> Our rationale to include Proposition 1 in the manuscript, in a similar manner to Proposition 1 in [1] is to highlight the worst case behavior of the SRDS algorithm and provide the guarantee on convergence, despite the triviality. We will be sure to include the cross-references to the appropriate equations in [1] and [2], especially in the larger context of fixed-point iteration for n-order triangular systems.
>
> > different spirit in parallelization with [1][2] ... which one is better from the theoretical aspects
>
>  This is a great question! Indeed, SRDS has a different spirit of parallelization compared to [1] and [2]. We like to view it as a multigrid/multiresolution method where we solve for the trajectory at a low resolution and use parallel high-resolution solves to correct the trajectory. This is in contrast to [1] and [2] which have the flavor of more traditional fixed point iteration/updates. From a theoretical point of view, it is actually unclear which is fundamentally better. Even if SRDS might seem slightly superior empirically today, all three (SRDS [1] and [2]) are still the first works in this area of parallel sampling, and it is entirely possible that one of them possesses superior theoretical properties/guarantees that can be exploited. As we briefly discussed in the future work section of the paper, convergence guarantees/properties for parareal have been analyzed only in a handful of special cases such as heat equations, and it would be a very interesting future direction to see if the diffusion ODE admits better convergence guarantees for parareal (in line with what is observed empirically).
>
> > SDE sampler
>
> Yes, as we mention in the paper, SRDS can indeed be readily applied to the SDE sampler. As seen in [1], we can just use the trick of pre-sampling the noise upfront, and the rest of the algorithm remains the same. Notably, it is even possible to ensure that the noise at the coarse and fine-time scales is aligned; however, this is not necessary due to the use of the predictor corrector updates in the parareal algorithm.
>
> ---
> References:
> - [1] Shih, Andy, et al. "Parallel sampling of diffusion models." Advances in Neural Information Processing Systems 36 (2023).
> - [2] Tang, Zhiwei, et al. "Accelerating parallel sampling of diffusion models." Forty-first International Conference on Machine Learning. 2024.

---

> > ### Comment · Reviewer_KhBd · 2024-08-07
> > **Thanks for your rebuttal**
> >
> > After careful consideration, I decided to raise my score to 6.

---

> > > ### Author Response · Authors · 2024-08-14
> > >
> > > Thank you for your response and for raising the score! We are glad that your concerns have been addressed.

---

### Official Review · Reviewer_pK7d · 2024-07-10

**Soundness:** 3
**Presentation:** 3
**Contribution:** 3
**Rating:** 6
**Confidence:** 3

**Summary:**

This paper proposes Self-Refining Diffusion Samplers (SRDS), which draws inspiration from the Parareal algorithm, aims to solve the reverse process accurately without retraining models, balance the tradeoff between sample quality and sampling speed. lower latency for requiring fewer sampling steps to reach convergence. The experiment section shows that the proposed algorithm speeds up the convergence without degrading sample quality in image generation tasks.

**Strengths:**

1. This paper is motivated by Parareal algorithm to speed up sampling process without lowering sample quality which seems to work well with image generation.
2. This paper provides theoretical analysis for sampling convergence and latency investigation.
3. This paper observes the pipelined SRDS, which further speeds up the sampling process.

**Weaknesses:**

1. In related work, the authors mentioned that ParaDiGMS proposed by Shih et al [1] is another parallel-in-time integration method, and this paper also works on image generation. In this case, I believe their results could be baselines that this paper should compare against.

[1] Andy Shih, Suneel Belkhale, Stefano Ermon, Dorsa Sadigh, and Nima Anari. Parallel sampling of diffusion models. Advances in Neural Information Processing Systems, 36, 2024.

**Questions:**

1. For the experiment section, are the numbers intentionally matched in ``FID Score`` in Table 1 and ``CLIP Score`` in Table 2? If that's true, it would be also interesting to see how the performance changes when you have different SRDS Iters and their corresponding Total Evals.

**Limitations:**

This paper discussed the limitations and pointed out the potential positive or negative social impacts will not be the direct consequence of this work in the checklist guidelines.

---

> ### Author Rebuttal · Authors · 2024-08-07
>
> We thank the reviewer for the review, and are happy to hear that the experimental results are convincing! We respond to the concerns below:
>
> > ParaDiGMs baseline
>
> Please refer to the combined response to all reviewers, where we clearly demonstrate the superiority of SRDS over ParaDiGMS. For instance, SRDS is up to 38% faster than ParaDiGMS for DDIM on identical hardware even when using very generous convergence thresholds for ParaDiGMS.
>
>
> > Are the numbers intentionally matched in FID Score in Table 1 and CLIP Score in Table 2
>
> Thanks for the question! Yes, the FID Score in Table 1 and CLIP Score in Table 2 are presented to show a tolerance/SRDS iterations so as to have no measurable degradation in sample quality. In the original version of the manuscript, we have included an ablation study for the number of SRDS iterations and the corresponding CLIP score in Figure 5; showcasing the capability of the model to achieve early convergence. For pixel space diffusion, we provide an ablation of tolerance threshold vs sample quality in Appendix C. We note that we use KID score instead of FID score due to computational budget constraints (KID is an unbiased estimator and requires much fewer samples than FID).

---

> > ### Comment · Reviewer_pK7d · 2024-08-11
> >
> > I have read through the responses and decide to increase the score from 5 to 6.

---

> > > ### Author Response · Authors · 2024-08-14
> > >
> > > Thank you for your response and for raising your score! We are glad to hear that your concerns have been addressed.

---

### Official Review · Reviewer_XqLH · 2024-07-10

**Soundness:** 2
**Presentation:** 1
**Contribution:** 2
**Rating:** 3
**Confidence:** 4

**Summary:**

Inspired by the parallel-in-time ODE integration literature, especially Parareal method, this paper introduces SRDS as a fast sampler for diffusion models enabling efficient parallelization. Experimental results demonstrate that the proposed SRDS reduces the number of steps required to synthesize samples.

**Strengths:**

It is interesting to use parallel sampling algorithms to accelerate the sampling of diffusion models, which will benefit practical applications such as real-time image editing.

**Weaknesses:**

My concerns are mostly about the limited experimental results:

1. This paper lacks of clear discussions about the differences between the proposed method and the previous work “23NeurIPS-Parallel sampling of diffusion models.”, and sufficient experimental evidence to demonstrate the effectiveness of the proposed method. Specifically, there is no quantitative comparison with the most related method ParaDiGMs. It seems that the existing method ParaDiGMs can provide better speedup according to their results.
2. This paper lacks of a sufficient comparison to the existing fast samplers (such as Heun, DEIS, DPM-Solver). In practice, we can already synthesize samples with those advanced samplers with only around 10 steps. While this paper only presents experiments on several hundreds or one thousand steps at Table 1, which makes the method proposed in this paper unappealing.
3. It is claimed in Lines 38-40 and 196-198 that SRDS outperforms ParaDiGMs in terms of memory requirement. However, there is also no quantitative support from experimental results.
4. It is claimed in Line 62-64 that SRDS is compatible with off-the-shelf solvers. However, the speedup is only shown on DDIM. The speedup offered by SRDS is only explicitly shown in Table 2, making it hard to fully assess the effectiveness of the proposed method.
5. There is no quantitative comparison between SRDS and pipelined SRDS.

**Questions:**

1. Could the pipelined SRDS be easily deployed in practice?
2. Why the SRDS Iters shown in Table 1 are not integers?
3. Why the first and second rows of Figure 6 are almost the same?
4. Minor points:

(1) In line 10 of Algorithm 1, $x_{i-1}^p$ is undefined for p == 1.

(2) It is redundant to use Figure 2, Figure 3b and Algorithm 1 to describe the same thing in the main text. They together occupy a whole page. Adding more experiments to support the effectiveness of the proposed method is more valuable.

**Limitations:**

Please refer to the weakness and question sections above. This manuscript needs an overhaul to make it meet the bar of a top conference.

---

> ### Author Rebuttal · Authors · 2024-08-07
>
> We thank the reviewer for the review, and appreciate that they recognize the practical applicability of our work! We respond to the concerns below:
>
> > ParaDiGMs baseline
>
> Please refer to the combined response to all reviewers, where we clearly demonstrate the superiority of SRDS over ParaDiGMS. For instance, SRDS is up to 38% faster than ParaDiGMS for DDIM on identical hardware even when using very generous convergence thresholds for ParaDiGMS.
>
> > Comparison to existing solvers
>
> We'd like to re-emphasize that SRDS provides an orthogonal improvement compared to the other lines of research on accelerating diffusion model sampling. In particular, while the main experiments (and writing) were focused on DDIM, SRDS is compatible with other solvers and they can be readily incorporated into SRDS to speed up diffusion sampling. We have empirically also evaluated that the SRDS method can be used with other solvers, such as DDPM (often requiring more steps than DDIM) and DPMSolver (often requiring fewer steps than DDIM). For both methods, SRDS maintains a speedup over the corresponding baseline serial implementation (~3.6x for 1000-step DDPM and ~1.5x for 25-step DPMSolver ).  Please see the combined response for a greater discussion of this extension.
>
> > SRDS outperforms ParaDiGMs in terms of memory requirement
>
> Apologies for the confusion. Yes, indeed ParaDiGMs has a higher memory requirement. Specifically, for a T step denoising process, ParaDiGMS needs to perform $T$ model evaluations in parallel with subsequent computations needing information about all previous evaluations, while SRDS only requires $\sqrt{T}$ parallel evaluations (which fits comfortably in GPU memory) and requires much lesser communication between GPUs. While the prohibitively large memory requirement can be combated with a sliding window method, the significantly larger communication overhead remains because at every step of Paradigms, an AllReduce over all devices must be performed in order to calculate updates to the sliding window. (For instance, even when ParaDiGMS reduces Eff. Serial Steps by 20x, the obtained speedup is only 3.4x). This is in contrast to the independent fine-solves in parareal that only need to transfer information for the coarse solve.
>
> Below, we demonstrate how the minimal memory and communication overhead of SRDS shines through as we are able to achieve better device utilization as we increase the number of available GPUs. The following experiment was performed on 40GB A100s and used a generous 1e-2 threshold for ParaDiGMS.
>
> |  |  | Serial | SRDS | SRDS | ParaDiGMS | ParaDiGMS |
> |---|---|:---:|:---:|:---:|:---:|:---:|
> |  | Devices | Model Evals | Eff Serial Evals | Time Per Sample | Eff. Serial Evals | Time Per Sample |
> | DDIM | 1 | 25 | 15 | **1.62** | 16 | 2.71 |
> | DDIM | 2 | 25 | 15 | **1.08** | 16 | 2.01 |
> | DDIM | 4 | 25 | 15 | **0.82** | 16 | 1.51 |
>
> In the table above, we can observe how SRDS has a significantly better time per sample than ParaDiGMS despite having a similar number of effective serial evaluations.
>
> > Pipelined SRDS
>
> We compare SRDS and pipelined SRDS in the overall response to all reviewers. We hope this provides more context and strengthens the empirical evaluation!
>
>
> > Could the pipelined SRDS be easily deployed in practice?
>
> Yes, the pipelined SRDS can be easily deployed in practice, though it does take minor adjustments to common diffusion modeling code in order to be efficiently implemented. For example, the StableDiffusion pipeline makes use of a scheduler that ahead-of-time sets the number of inference steps (as a class variable). Adjusting this parameter to be modified at runtime is required for pipelining. Additionally, a framework is required to coordinate computation and data transfers; our implementations make use of torch.multiprocessing and the queue organization to coordinate the launches of the different nodes.
>
> While extracting close to optimal bonus speedup (2x) from pipelining might require considerable engineering effort, we implemented a suboptimal version of pipelined SRDS and already achieve substantial speedups compared to non-pipelined SRDS and more importantly baselines such as ParaDiGMS. Please refer to the combined response for quantitative results on the same.
>
> > Why the SRDS Iters shown in Table 1 are not integers?
>
> In Table 1, for pixel-based diffusion, we used a very tight/conservative threshold on the convergence of the output in order to define early convergence. We then measure the average number of SRDS iterations required until convergence across 5000 samples, which ends up not necessarily being an integer. This contrasts from Table 2 (Stable Diffusion) which caps the number of iterations with no measurable degradation in sample quality. We will make this more clear in the revised manuscript text, as well as add an equivalent version of Table 1 that includes a capped number of iterations in order to obtain a fairer comparison to baselines (e.g. ParaDiGMS) and demonstrates much greater speedups without loss of sample quality. In fact, Figure 7 in Appendix C already provides support for the fact that even for pixel-based diffusion models, we can similarly limit the number of SRDS iterations to 1 or 2 without degradation in sample quality, while achieving much greater speedups than reported in Table 1.
>
> > Why the first and second rows of Figure 6 are almost the same?
>
> The first and second rows of Figure 6 seek to highlight the early convergence of the SRDS algorithm to the output of sequential sampling. For a visualization of the ‘iterative refinement’ through rounds of parareal iterations, we direct the reviewer to Figure 1.

---

### Official Review · Reviewer_MmLD · 2024-07-11

**Soundness:** 3
**Presentation:** 3
**Contribution:** 3
**Rating:** 6
**Confidence:** 4

**Summary:**

This work proposes SRDS, a sampler for diffusion models that applies the Parareal algorithm, to reduce the overall sampling latency by introducing extra but parallelizable network evaluations compared to the fully sequential fashion. With higher device utilization or device parallelism through batched inference and pipelining, SRDS successfully reduces the overall sampling latency of existing diffusion models without sacrificing the sample quality.

**Strengths:**

- This paper is well written.
- The proposed SRDS sampling pipeline is flexible and extendable, providing a neat baseline and rich directions for future technical improvements.
- The experiment successfully demonstrates SRDS's practical value of reducing the sampling latency compared to full sequential sampling.
- The ablation studies provide a nice practical guidance for SRDS.

**Weaknesses:**

- As the authors addressed in the limitation part, SRDS relies on parallelizing extra computations in exchange for a reduction in total latency, which may not be applicable to some scenario that performs full batch sequential sampling. However, I believe SRDS could still work in a fairly broad range of cases.
- The important ParaDiGMs baseline is not compared in the experiment part.

### Minor
- The meaning of the abbreviation "IVP" in line `1:` of Algorithm.1  is unclear.

**Questions:**

- In the current SRDS pipeline, the intermediate results of the fine-grained solver are only used once. Do you have any thoughts on utilizing them in subsequent steps for better sample quality or faster convergence?

**Limitations:**

The authors have adequately addressed the limitations of their work.

---

> ### Author Rebuttal · Authors · 2024-08-07
>
> We thank the reviewer for the review and are glad to hear the reviewer’s agreement of the broad applicability of SRDS! We respond to the concerns below:
>
> > ParaDiGMs baseline
>
> Please refer to the combined response to all reviewers, where we clearly demonstrate the superiority of SRDS over ParaDiGMS. For instance, SRDS is up to 38% faster than ParaDiGMS for DDIM on identical hardware even when using very generous convergence thresholds for ParaDiGMS. We hope that addition of these results, as well as the comparisons between different samplers, helps provide more context to the impact of SRDS in diffusion model sampling.
>
> > In the current SRDS pipeline, the intermediate results of the fine-grained solver are only used once. Do you have any thoughts on utilizing them in subsequent steps for better sample quality or faster convergence?
>
> We appreciate the question about only using the intermediate results of the fine-grained solver once, as it highlights that there is a rich body of work in parallel-in-time integration methods that may also be applicable to parallelizing the sampling of diffusion models. In particular, the intermediate re-use idea appears similar to PITA [1]; however, the inversion required in the calculation of the projection matrix makes it more complicated and less amenable to the high-dimensional outputs common to diffusion models. There may be room for such methods in latent diffusion; though have not yet explored such an approach due to the complexity.
>
> There may be more room for simpler ideas in diffusion sampling, as the approximation to the score function may be inaccurate and simple averaging in re-use may provide a better approximation of the true score. These have not been explored as the main goal of the paper was to ensure that parallelized sampling was finding the solution to the ODE defined by the network. This would certainly be an interesting future direction to explore theoretically and empirically as an extension to SRDS!
>
> > The meaning of the abbreviation "IVP" in line 1: of Algorithm.1 is unclear.
>
> Thanks for pointing this out. We will update the manuscript to clarify that it refers to ‘Initial Value Problem’.
>
>
> ---
>
>
> References:
>
> - [1] C. Farhat and M. Chandesris, “Time-decomposed parallel time-integrators: theory and feasibility studies for fluid, structure, and fluid-structure applications,” International Journal for Numerical Methods in Engineering, vol. 58, no. 9, 2003, doi: 10.1002/nme.860. [Online].

---

> > ### Comment · Reviewer_MmLD · 2024-08-12
> >
> > I would like to thank the authors for the rebuttal, especially the additional results. I will keep my current recommendation unchanged.

---

> > > ### Author Response · Authors · 2024-08-14
> > >
> > > Thank you for your response! We are pleased to hear that your concerns have been addressed.

---

### Official Review · Reviewer_dCML · 2024-07-11

**Soundness:** 3
**Presentation:** 3
**Contribution:** 2
**Rating:** 6
**Confidence:** 3

**Summary:**

The paper presents a new approach to speed up (improve latency)  the generation of samples from diffusion models. The approach is orthogonal to many other approaches present in the literature for the same task. It leverages Parareal algorithm for the task by getting a quick coarse approximate of the sample and then refining it iteratively in parallel, thus reducing the latency while maintaining sample quality. The authors present results for pre-trained pixel space diffusion models  and Stable Diffusion and find upto 2x speed ups for the latter.

**Strengths:**

Improving the latency of diffusion models is an active and important area of research. The paper presents a new strategy that relies on leveraging parallel computation of modern hardware. This approach can in theory be combined with other strategies to lead to further improvement, and does not require re-training. The presented approach seems to have guarantees on the quality of the solution and some control on trading off the speed and quality. There is also thought given to efficient batching and pipelining tasks for further potential improvements, which is good. Overall, the paper is also well written and easy to follow.

**Weaknesses:**

The main weakness of the paper is lack of strong and diverse results. The authors only test one  diffusion model task. Furthermore the best speed up is only 2.3x, while another experiment results in 0.7x speed up which is concerning.
The authors argue on theoretical grounds that this approach can be combined with other approaches to reduce latency of diffusion models. This is fair, but they do not show any empirical results around it.
Finally, the approach relies completely on leveraging parallel refinement and the actual number of model evaluations are much larger (often by a factor of 3x) which can limit the applicability in some cases.

**Questions:**

- The authors claim that the worst case sampling latency is no worse than generating a sample through sequential sampling. Yet one of their results (last on in Table 2) has speed up of 0.7x. Is this all due to GPU overhead?

- Is is possible to present empirical results with pipeline parallelism? It seems like the authors expect 2x gains with it, but I assume it will require some computational overhead and based on the previous question, the degradation can be non-trivial.

- While I understand the claim regarding this approach being orthogonal to other works reducing, showing some experiments around the same will make for a stronger paper.

**Limitations:**

There are a few limitations discussed towards the end of the paper.

---

> ### Author Rebuttal · Authors · 2024-08-07
>
> We thank the reviewer for the review, and appreciate that they find the paper well written and easy to follow! We respond to the remaining concerns below:
>
> > The authors argue on theoretical grounds that this approach can be combined with other approaches to reduce latency of diffusion models. This is fair, but they do not show any empirical results around it.
>
> In order to empirically demonstrate that the SRDS method can be used in conjunction with other approaches to reduce the latency of diffusion model sampling, we have extended beyond the base DDIM solver primarily used in the paper. We tested SRDS both in conjunction with DDPM (which takes relatively more steps than DDIM) and DPMSolver (which can reduce the number of steps beyond DDIM). For both methods, SRDS maintains a speedup over the corresponding baseline serial implementation (~3.6x for 1000-step DDPM and ~2.9x for 200-step DPMSolver ). In this way, we show that SRDS for example could be combined with the improvement from DDPM → DDIM or DDIM → DPMSolver in order to orthogonally improve latencies. Please refer to the combined response to all reviewers for more details on the same.
>
> > The approach relies completely on leveraging parallel refinement and the actual number of model evaluations are much larger (often by a factor of 3x) which can limit the applicability in some cases.
>
> Yes, indeed there is a tradeoff: as we note in the introduction section, SRDS provides speedups in sampling at the cost of potentially additional parallel compute. While this might limit applicability in some cases (such as compute-bound, batched inference workloads), we would like to emphasize that this tradeoff enables the diffusion models for many other use cases such as real-time image or music editing and trajectory planning in robotics. In fact, there is an increasing trend of diffusion models run locally where small enough models or strong enough hardware can support the batching described in this paper. In such scenarios, strict latency requirements may justify the tradeoff of additional compute vs wall-clock times. Moreover, in a number of cases, we have found that SRDS provides reasonable predictions within a single Parareal iteration; here, the total number of model evaluations is only slightly larger than the serial approach (increasing from $n$ to $ n + 2\sqrt{n}$). Lastly, it is also worth noting that many users are often agnostic to inference time GPU compute costs as they are orders of magnitude lower than training compute costs anyway.
>
>
> > Yet one of their results (last on in Table 2) has speed up of 0.7x. Is this all due to GPU overhead?
>
> No, while there is certainly some loss due to GPU overhead, the <1.0x speedup is primarily due to the lack of pipeline parallelism (meaning that coarse and fine solves are performed serially rather than in parallel). Note that the worst case sampling latency being no worse than sequential sampling only holds for the pipelined version. Without pipelining, in the worst case, the speed up is 0.5x, and the 0.7x speaks to the early termination/convergence of SRDS. Please refer to Figure 4 for an illustration that clarifies this.
>
> > Is is possible to present empirical results with pipeline parallelism? It seems like the authors expect 2x gains with it, but I assume it will require some computational overhead and based on the previous question, the degradation can be non-trivial.
>
> Please refer to the combined response to all reviewers for a broad discussion. While achieving close to 2x bonus gains with pipeline parallelism requires considerable engineering effort, we implemented a somewhat suboptimal version of pipelined SRDS, and this already showcases significant speedups compared to both non-pipelined SRDS (~20% speedup) and more importantly baselines such as ParaDiGMS (up to 38% faster than ParaDiGMS for DDIM on identical hardware even when using very generous convergence thresholds for ParaDiGMS; even faster when using stricter thresholds). The less than optimal speedups stems from GPU and scheduling overhead.
>
> > While I understand the claim regarding this approach being orthogonal to other works reducing, showing some experiments around the same will make for a stronger paper.
>
> We hope the discussion/experiments in the common response on readily incorporating other solvers such as DDPM and DPMSolver into SRDS helps make this a stronger paper and provides context to how SRDS can be used in conjunction with other improvements in diffusion modeling!

---

> > ### Comment · Reviewer_dCML · 2024-08-09
> > **Response to rebuttal**
> >
> > I thank the authors for their responses and new experiments. I am satisfied with their rebuttal and will maintain my score.

---

> > > ### Author Response · Authors · 2024-08-14
> > >
> > > Thank you for your response! We are glad to hear that your concerns have been addressed.

---

### Author Rebuttal · Authors · 2024-08-07

We thank the reviewers for their thoughtful reviews! We are thrilled to hear that the reviewers find that our paper is “well written and easy to follow”, “demonstrates a very interesting direction”, provides “a neat baseline and rich directions for future technical improvements”, and “will benefit practical applications such as real-time image editing”.

Here, we answer common questions:
- Empirical comparison to ParaDiGMS baseline: Up to 38% faster than ParaDiGMS for DDIM on identical hardware even when using very generous convergence thresholds for ParaDiGMS; even faster when using stricter thresholds!
- Quantitative evaluation of pipelined SRDS: Up to 20% faster than non-pipelined SRDS even with a suboptimal implementation!
- Incorporation of other samplers: SRDS with DDPM and DPMSolver yield similar speedups of up to ~3.6x over sequential!

Other questions are answered in individual responses to reviewers. Further, we’ll include these results/clarifications in our revised manuscript.

# Comparison to Baselines
We demonstrate the superiority of SRDS to baselines ParaDiGMS [1] and ParaTAA [2]. We especially thank reviewer KhBd for bringing [2] to our attention. [2] takes a different approach to accelerating parallel sampling (solving triangular nonlinear systems using special techniques) and we'll include a thorough discussion of it in the final manuscript. Unfortunately, as this is a new paper (ICML 2024), we were unaware of it while writing our paper, and consequently we're unable to provide a comprehensive empirical comparison with [2] with identical hardware at the moment. Nonetheless, for now, we demonstrate the high-level superiority of SRDS solely by using the results published by the authors in [1] (Table 5) and [2] (Table 1).

In the table below, we show that SRDS offers better wall-clock speedups (over sequential) in sample generation time for StableDiffusion when compared to [1] and [2]. We clarify that the reported speedup for each method is w.r.t sequential solve on the same machine that the corresponding parallel method was evaluated. Our results below are particularly impressive given that the authors of [1] used 8x 80GB A100s for the evaluation and the authors of [2] used 8x 80GB A800 for the same, while we (SRDS) only used 4x 40GB A100 for the evaluation due to computational constraints. (When interpreting, recall that a sequential solve isn't compute/memory bound and doesn’t benefit significantly from more GPU compute, whereas the parallel methods certainly do!) We'd also like to highlight the superiority of SRDS over the baselines in the regime of small number of denoising steps (25) as particularly impactful.
||Denoising Steps|ParaDiGMS|ParaTAA|Pipelined SRDS|
|---|:---:|:---:|:---:|:---:|
|DDIM|100|2.5x|1.92x|**2.73x**|
|DDIM|25|1.0x|1.17x|**1.72x**|

For the main baseline ParaDiGMS that we consider in our paper, we also performed more extensive evaluation to evaluate both methods on equal hardware to more clearly demonstrate the benefits of SRDS. Below, we demonstrate that SRDS consistently beats ParaDiGMS on wallclock speedups. Though the ParaDiGMS paper uses a convergence threshold of 1e-3, we show that SRDS can provide impressive speedups even when compared to significantly relaxed ParaDiGMS thresholds of 1e-1. The following StableDiffusion experiments are performed on identical machines (4 40GB A100 GPUs) for a fair comparison.
||Serial|Serial|Pipelined SRDS|ParaDiGMS|ParaDiGMS|
|---|:---:|:---:|:---:|:---:|:---:|
||Model Evals|Time Per Sample|Time Per Sample|Threshold|Time Per Sample|
|DDIM|961|44.88|**10.31 (4.3x)**|1e-3|275.29|
|||||1e-2|20.48|
|||||1e-1|14.30|
|DDIM|196|9.17|**2.85 (3.2x)**|1e-3|29.45|
|||||1e-2|5.08|
|||||1e-1|3.42|
|DDIM|25|1.18|**0.69 (1.7x)**|1e-3|1.98|
|||||1e-2|1.51|
|||||1e-1|0.77|

# Pipelined SRDS
We demonstrate that pipelining can indeed further speed up SRDS. We implemented a slightly suboptimal version of pipelined SRDS for StableDiffusion below and already observe significant speedups; with some more engineering effort, we can further push towards extracting the full potential of pipelining. However, we believe that because this already beats the baselines, this sufficiently demonstrates the benefits of SRDS.

Our implementation is suboptimal due to using a single device for coordinating pipeline parallelism and device transfers (artifact from torch.multiprocessing). A better approach would use ring-like communication between devices instead of relying on a coordinator.
||Serial|SRDS|SRDS|Pipelined SRDS|Pipelined SRDS|
|---|:---:|:---:|:---:|:---:|:---:|
||Model Evals|Eff Serial Evals|Time Per Sample|Eff Serial Evals|Time Per Sample|
|DDIM|961|93|12.30|63|**10.31**|
|DDIM|196|42|3.30|27|**2.85**|
|DDIM|25|15|0.82|9|**0.69**|

# Incorporation of Other Solvers
As stated in our paper, we'd like to emphasize that SRDS provides an orthogonal improvement compared to the other lines of research on accelerating diffusion model sampling. In particular, while the main experiments (and writing) were focused on DDIM, SRDS is compatible with other solvers and they can be readily incorporated into SRDS to speed up diffusion sampling. For example, SRDS is directly compatible with other solvers such as DDPM (often requiring more steps than DDIM) and DPMSolver (often requiring fewer steps than DDIM) and can efficiently accelerate sampling in both cases, as shown with StableDiffusion below.
||Sequential|Sequential|SRDS|SRDS|SRDS|
|---|:---:|:---:|:---:|:---:|:---:|
||Model Evals|Time Per Sample|Eff Serial Evals|Time Per Sample|Speedup|
|DDPM|961|44.68|93|12.30|**3.63x**|
|DDPM|196|9.03|42|3.26|**2.76x**|
|DPMSolver|196|10.30|42|3.49|**2.95x**|
|DPMSolver|25|1.31|15|0.88|**1.48x**|
|DDIM|196|9.17|42|3.30|**2.77x**|
|DDIM|25|1.18|15|0.82|**1.43x**|

We also highlight that our Diffusers-compatible implementation needs only minor changes to solver arguments, indicating that SRDS can likely be easily extended to future community-developed methods.

---

### Author Response · Authors · 2024-08-07
**Thanks for the reviews!**

We sincerely hope that the additional experimental results/clarifications in the rebuttal clearly demonstrate/validate the usefulness of our proposed method (SRDS), thereby sufficiently answering all the reviewers’ concerns and significantly improving the strength of the paper. We once again thank the reviewers for their time and feedback, and we are happy to answer any further questions that may remain in the discussion period!

---

Author Rebuttal References:
- [1] Shih, Andy, et al. "Parallel sampling of diffusion models." Advances in Neural Information Processing Systems 36 (2023).
- [2] Tang, Zhiwei, et al. "Accelerating parallel sampling of diffusion models." Forty-first International Conference on Machine Learning. 2024.

---

### Decision · Program_Chairs · 2024-09-25

**Decision:**

Accept (poster)

**Comment:**

This paper addresses the computational complexity of Diffusion Models, and seek to overpower the slow sequential evaluations which typically drive  diffusion model generation.  Specifically they propose Self-Refining Diffusion Samplers (SRDS) to improve the speed at a cost of parallel compute while preserving the sample quality.  This is in contrast to prior attempts at the problem toying with discretization. Convergence results on an ODE have been shown and comparative experiments were carried out.
 While all 4  reviews out of 5 rated the paper as a "weak accept", and one as reject,  several critical comments were made, although most were successfully rebutted by the authors to the reviewers' satisfaction.  One significant one pertained to the proposed Parallel computational  non-novelty  with  respect to some specific recent references cited refs [1]& [2] by reviewer kHBd.
The authors have responded to the reviewer's satisfaction that in contrast to the multigrid/multiresolution flavor, the prior cited refs.  used classical  fixed point iteration/updates.
The "Reject" review also raised several questions and some due to confusion and others were in the end rebutted to the reviewer's agreement who nevertheless kept their reject intact. Judging by the comments, I believe that the reviewer is a little confused,
I think sufficient experimental evidence was also provided as attested by the other 4 reviewers.

.